# Longitudinal Changes in the Parenting Stress of Mothers of Children with Cerebral Palsy and Their Relationship with Children’s Gross Motor Function System Levels

**DOI:** 10.3390/healthcare11091317

**Published:** 2023-05-04

**Authors:** Eun-Young Park

**Affiliations:** Department of Secondary Special Education, Jeonju University, Jeonju 55069, Republic of Korea; eunyoung@jj.ac.kr; Tel.: +82-63-220-3186

**Keywords:** mothers, children with cerebral palsy, gross motor classification system, parenting stress, longitudinal study, latent growth model

## Abstract

Stress caused by children with disabilities harms the growth and development of children and their families. The present study aimed to investigate whether parenting stress of mothers of children with cerebral palsy changes and the relationship between children’s gross motor function level and changes in parenting stress. Data were collected from 162 children with cerebral palsy and their mothers over a 2-year period at three time points. Gross motor function and parenting stress were assessed using the Gross Motor Function Classification System and the Korean version of the Parenting Stress Index—Short Form. Linear latent growth curve models of Wave 2 and Wave 3 variation in parenting stress were constructed. The relationship between Gross Motor Function Classification System level and parenting stress was evaluated via latent growth modeling. The results showed that the linear variation models reflected the time evolution of parenting stress. There were individual differences in parenting stress at the initial level and no individual differences in changes in parenting stress. The relationship between the initial level and the change rate of parenting stress was not significant. Higher initial values of the Gross Motor Function Classification System level were associated with higher initial values of parenting stress, but not related to a change in parenting stress. The results showed that parenting stress of mothers with children with cerebral palsy decreased over time and that Gross Motor Function Classification System level was related to parenting stress level. Based on these findings, directions for further research are put forth.

## 1. Introduction

Parents experience various stressors while raising children, collectively referred to as parenting stress [1]. Numerous studies have demonstrated that the parenting stress of mothers who care for children with disabilities is higher than that of mothers of children with normal development [2,3,4]. Factors that increase parenting stress include the child’s age [5], degree of disability [6,7], and problem behavior [8]. Factors that decrease parenting stress include social support [9,10], social network [11], and coping strategies [12].

Continuous parenting stress is highly likely to cause depressive symptoms due to an increase in negative emotions, such as anxiety, nervousness, feelings of inferiority, and decreased confidence in parenting [13,14,15]. Additionally, parenting stress is a factor that hinders child development by negatively affecting parent–child interactions [16]. When mothers treat their children affectionately and acceptably by regulating their emotions, children develop high social skills [17]. Increased perceived stress in mothers indicates a more coercive and rejecting parenting behavior toward their child. In response, the child’s self-esteem is lowered [18] and anxiety and depression can develop [19], impairing social skill acquisition [1].

Research on parents of children with cerebral palsy (CP) generally indicates that their levels of parenting stress are higher than those of parents with able-bodied children [20]. Although it is not clear why stress levels vary among parents, a recent review of factors involved in the adaptive process has examined many child- and parent-related factors and various sources of contextual support [21]. Studies on disabilities among children that affect parenting stress have continued to be published, reporting the impact of differing severity of disability or gross functional level of children with CP. Specifically, gross motor function was reported to be a variable that significantly elevated parenting stress [22]; however, other findings indicated that the degree of motor function impairment did not correlate with parenting stress [7].

Boss [23] described parenting stress as the tension and pressure within the family system that individuals and families inevitably endure in the process of development and change over time; it is the degree of difficulty or burden perceived by parents due to child rearing. As such, studies on parenting stress and children’s development show that parenting stress is not a temporary phenomenon, but one that is constant throughout the child-rearing process, with long-term effects on parental depression and child development. Crnic et al. [16] revealed that mothers’ parenting stress is a constant characteristic that does not change significantly and is a major variable that can directly or indirectly affect maladjustment during infant growth in USA.

There is a need to empirically investigate the changes in the parenting stress of mothers of children with CP, the variables that affect the pattern of such changes, and their evolution with the passage of time. A cross-sectional approach to parenting stress in mothers of children with CP has yet to be attempted, and longitudinal studies focusing on the pattern of change or influencing factors are rare. Therefore, this study intended to analyze changes in parenting stress in mothers of children with CP over time and how they are affected by children’s gross motor function level using a longitudinal approach.

The research questions for this study are as follows: First, does the parenting stress of mothers of children with CP change? Second, does the level of gross motor function in children with CP affect their mothers’ parenting stress?

## 2. Materials and Methods

### 2.1. Participants

The participants comprised 162 children with CP receiving rehabilitation treatment at a hospital or community welfare center or attending a school for children with physical disabilities, and their mothers. All mothers were primary caregivers. Sampling was performed across the country, with the exception of Jeju Island because there was no expression of intention to participate. The inclusion criteria were (1) children with a medical CP diagnosis and (2) children ranging from 3 to 15 years of age. The participants were mothers who responded thrice during the two-year study period. This study was approved by the Research Ethics Board of Jeonju University on 14 November 2013 (ethical approval code number: Jeonju University IRB-1041042-2013-1).

### 2.2. Measures

The Gross Motor Function Classification System (GMFCS) was used to evaluate the children’s gross motor function. Motor skills were assessed at five levels, with level 1 indicating being able to walk without any restrictions, level 2 indicating being able to walk with restrictions, and level 3 indicating being able to walk without trunk support using canes, crutches, or walkers. Level 4 indicates that a child’s walking is limited, but they are able to move independently using an electric wheelchair or other means of transportation, and level 5 indicates that mobility is severely limited even with assistive devices [24]. Park [25] reported that the coefficients ranged from 0.690 to 0.789, and the GMFCS remained stable in children with CP aged 2 to 12 years within South Korea.

The Korean Parent Stress Index—Short Form (K-PSI-SF) was used to evaluate the mothers’ parenting stress. The K-PSI-SF is a tool for the identification and diagnosis of parenting-related stress, and it is designed to measure the relative stress level in a parent–child relationship. It can be used by parents with children over one month. It consists of three domains, including parent distress (PD), parent–child dysfunctional interaction (PCDI), and difficult child characteristics (DCC). Each domain consists of twelve questionnaires. The validity and reliability of K-PSI-SF had been confirmed through a Rasch analysis of mothers having children with CP; the internal consistency (Cronbach’s α) of the entire test was reported to be 0.92 [26].

### 2.3. Procedure

To investigate changes in the parenting stress of mothers raising children with CP, 162 children according to the selection criteria were sampled, and data were collected in the first year. This study was approved by the Jeonju University Research Ethics Committee (Jeonju University IRB-1041042-2013-1). In Wave 1, data on the GMFCS level and parenting stress were investigated; in Waves 2 and 3, only data on parenting stress were investigated. The level of the GMFCS was evaluated by a physical therapist who had treated the child for more than six months. Participants in Wave 1 also participated in Waves 2 and 3. The mothers responded to the K-PSI-SF through a face-to-face survey. When the mothers visited the hospital, community welfare center, or school for children with physical disabilities, they received the questionnaire. After the third evaluation, it was investigated whether changes in parenting stress and the level of the GMFCS in Wave 1 affected parenting stress in subsequent years using a latent growth curve model.

### 2.4. Statistical Analysis

A latent growth model was applied to find out how the parenting stress of the participating mothers changed over time. Covariance structure analysis was used to apply and analyze the latent growth curve modeling to actual data, which necessitated a sample size of at least 150 [27]. The participants comprised the mothers of 162 children with CP receiving rehabilitation treatment at hospitals or community welfare centers, which met the sample size for the latent growth curve modeling.

The basic linear change model, the change model in Wave 2, and the change model in Wave 3 were first verified using the parenting stress evaluation data from the first to the third measurement. The suitability of linear change in parenting stress was tested by comparing the fitness indices of the change models. In this study, the comparative fit index (CFI), the non-normalized fit index (NNFI or Tucker–Lewis index: TLI), and the normed fit index (NFI), which are relative fit indices, were used. If the fit indices are above 0.90, the model is considered to have good fit [28]. Root mean square error of approximation (RMSEA) values of 0.06–0.08 indicate a reasonable fit. Researchers estimating models from data do not trust χ^2^ because the power of the χ^2^ test is lacking [29]. Therefore, the χ^2^ test was not used to examine the fit indices in this study.

After testing the change models of parenting stress, the relationship between the change trajectory of parenting stress and the GMFCS level was investigated using multivariate latent growth modeling. For descriptive statistical analysis, such as mean, standard deviation, and correlation, SPSS 26.0 (NY, USA) was used; for latent growth modeling analysis, AMOS 26.0 (NY, USA) was used.

## 3. Results

### 3.1. Descriptive Statistics

The mean age of the children with CP was 9.6 years (SD = 4.69) at baseline. The general characteristics of the participants are shown in Table 1. There were 90 boys (55.6%) and 72 girls (44.4%). The children with CP were classified using the GMFCS: 21 (13.0%) were classified into level 1, 25 (15.4%) into level 2, 19 (11.7%) into level 3, 21 (13.0%) into level 4, and 76 (46.9%) into level 5. There were significant differences in the GMFCS level frequency. The most frequent age range of the mothers of children with CP was the 40–49 age group (*n* = 120, 74.1%), and the least frequent age range was over 50 years of age (*n* = 9, 5.6%). University graduation was the most common educational level in the mothers of children with CP (*n* = 92, 56.8%) followed by high school graduation (*n* = 44, 27.2%). Most mothers of children with CP were unemployed (*n* = 107), with 51 employed mothers. Furthermore, there were significant differences in the frequencies based on the mothers’ general characteristics, such as age, education level, and employment.

The mean parenting stress values according to the sub-categories and GMFCS levels are shown in Table 2. According to the sub-categories, the values for parental distress and difficult child characteristics decrease. Regarding the parent–child dysfunction interaction domain, the mean of the first measurement is highest, and the mean of the third measurement is higher than that in the second measurement. There is no specific pattern in parenting stress change according to the GMFCS levels. The mean parenting stress is 97.58 (SD = 19.03) in the first measurement, 94.19 (SD = 18.03) in the second measurement, and 93.87 (SD = 18.34) in the third measurement (Table 2).

The intercorrelations among the study variables are presented in Table 3. The GMFCS level and parenting stress in the first year are significantly related (*p* < 0.05). The coefficient is 0.623 between the first measurement of parenting stress and the second measurement. The coefficient is 0.606 between the first and third measurements. The coefficient is 0.650 between the second and third measurements. The correlation coefficient between children’s age and the GMFCS level is 0.275 (*p* < 0.01).

### 3.2. Analysis of Change in Parenting Stress and the GMFCS Level

The results regarding the models’ quality of fit are presented in Table 4. Upon a comprehensive examination of the value of χ^2^, RMSEA, CFI, and TLI, it is observed that the linear change model is the best fit for analyzing parenting stress.

Table 5 shows the changes in parenting stress based on the linear change model. The initial value for parenting stress is 95.883 (*p* < 0.001) on average, with a variance of 209.505 (*p* < 0.001), and the rate of change is on average −2.552 (*p* < 0.05), with a variance of −14.908. The mean and variance of the initial values are significant, indicating that individual differences exist in parenting stress. The average of the significant rate of change suggests that parenting stress has changed significantly over two years, and the non-significant variance means that the trajectory of change over two years of parenting stress has no significant individual difference for the children with CP. The significance of the covariance reflects the relationship between the initial value and the rate of change. In the case of parenting stress, the covariance is not statistically significant.

### 3.3. The Relationship between Evolution of Parenting Stress and GMFCS Level

The results regarding the relationship between the evolution of parenting stress and GMFCS level are presented in Table 6 and Figure 1.

The suitability of the model is verified as the value of χ^2^ is found to be 18.186 (degree of freedom = 10), and the values of NFI, CFI, TLI, and RMSEA are 0.914, 0.910, 0.957, and 0.071, respectively. The results show that the higher the initial values of GMFCS level, the higher the initial values of parenting stress (B = 1.994, *p* < 0.05). There is no relation between changes in parenting stress and the GMFCS level of children with CP.

## 4. Discussion

Mothers experience parenting stress during the performance of parenting roles and throughout the developmental process of the family [30]. Previous studies on mothers’ parenting stress have reported that this stress changes gradually [31,32], but studies on such changes in mothers of children with CP are few. Parenting stress weakens mothers’ interest in child rearing, leading to dysfunctional parenting behavior [33]. The focus of this discussion is to provide recommendations for practices and future research directions regarding the major findings.

First, it was suggested that the parenting stress of mothers of children with CP decreased over time. To test the change trajectory of parenting stress, linear first-year and second-year change models were set as competition models and analyzed using a latent growth model. It became evident that the linear change model best fitted the analysis, indicating that parenting stress changed over time in mothers of children with CP. The average change rate was negatively significant, indicating that parenting stress decreased linearly over time. The parenting stress level of mothers in Korea is high due to low awareness and the burden of raising children during the early stages of childbirth [34]. However, over time, parenting stress gradually decreases due to reduced childrearing burden, increased experience of parenting, and emotional stability offered through the parent–child interactions. To provide the clinical meanings of the results, the minimal clinical important differences (MCID) were calculated using Cohen’s *d* [35]. The Cohen’s *d* between Wave 1 and Wave 2 was 0.02; the value was 0.02 between Wave 2 and 3; and the value was 3 between Wave 1 and Wave 3. Although the MCIDs across the measurement time were small, decreasing stress in mothers of children with CP might indicate the mothers’ adaptability to child rearing. The initial value of the parenting stress variable and the variance of the initial value were found to be statistically significant, implying that there were individual differences in the level of parenting stress at each time point. The rate of change was significant and the variance of the rate of change was not significant, indicating that parenting stress decreased significantly over time and that there was no difference between individuals with different GMFCS levels.

Previous studies have reported that the parenting stress of parents of infants initially increases and subsequently decreases after a certain age. Mulsow and Caldera [30] measured changes in parenting stress in 164 mothers and infants using random sampling and reported that the mothers’ parenting stress initially increased and then decreased between two and three years of age in the USA. Williford and Calkins [31] measured changes in parenting stress among 430 mothers of children between the ages of two and five and reported that the mothers’ parenting stress decreased after two years of age in the USA. A longitudinal study on parenting stress for young, low-income, African American first-time mothers (*n* = 120) was conducted by Chang and Fine [36], which showed that mothers’ stress gradually decreased with their children’s increasing age. The pattern of parenting stress reduction in mothers of children with CP is consistent with the results of studies showing that parenting stress reduces after infancy. This is likely because the children in this study were three years of age or older. Therefore, in future research, it is necessary to conduct a study on the changing patterns of parenting stress for mothers of children with CP in infancy.

Despite high stress levels, not all caregivers of children with disabilities have difficulties adjusting [5]. While some families have difficulty managing the circumstances associated with the process of rearing a child with a disability, some families are successful in adjusting [37]. In studying human adaptation, instead of focusing on the negative effects of adversity on humans or on vulnerabilities that are prone to negative states, the strengths of the participants and their tendency to change perspectives in the direction of strengthening their resources should be recognized [38]. Adaptive flexibility or resilience is a concept that best expresses the ability to successfully recover by reducing negative impacts and creating positive changes, such as growth, despite the crises and stressors that are inevitably encountered in life [39]. Until now, studies on the parenting stress of parents of children with disabilities have reported findings that show that the parenting stress of caregivers of children with disabilities is higher than that of caregivers of healthy children [2,3]. Therefore, in future studies, it is necessary to study the factors affecting the reduction in mothers’ parenting stress for children with CP.

Second, it was found that the level of gross motor function of the children with CP had a significant effect on the initial value of their mother’s parenting stress. The functional level of children with CP is regarded as a major factor influencing parenting stress [22]. Studies on the influence of the degree of disability or functional level of children with CP have uncovered different findings, but most studies agree that the more severe the disability of the children with CP, the higher the parental stress [22,40]. The effect of children’s GMFCS level on the initial parenting stress value in this study was in line with the findings of previous studies. However, there is a difference in the results of this study; the children’s GMFCS level did not have an effect on the change in the parenting stress of their mothers. The mothers’ parenting stress decreased regardless of their child’s GMFCS level. These results are probably related to the stability of GMFCS levels in children with CP. Stability of GMFCS levels in children with CP has already been reported [41,42]. While the GMFCS level did not change over time, the mothers’ parenting stress decreased. This implies that there were factors that mitigated parenting stress. The results of this study, which show that parenting stress decreases gradually in mothers of children with CP, suggests that it is necessary to investigate factors that affect the reduction in parenting stress.

There were some limitations in this study. First, there might be potential mediating variables related to parenting that might have affected the results. In future studies, it is necessary to investigate and verify mediating variables, such as numbers of children in the same household and marital status. This study investigated the relation between changes in parenting stress and GMFCS levels of children with CP in only mothers. Although mothers are the primary caregivers for children with CP, studies should be conducted to find out the caregiving burden of other caregivers, such as fathers and older siblings. Additionally, a study should be conducted to test how their caregiving burden affects family functioning. As the collection of the data used in this study began in 2007 and ended in 2009, potential effects of time should be considered while interpreting the findings. Since there is a possibility that factors related to parenting style and methods have changed after the data collection, an additional longitudinal study on the parenting stress of mothers of children with CP could provide data for comparison with this study’s results.

## 5. Conclusions

This study is significant as it reports the results of longitudinal changes in parenting stress in mothers of children with CP. The results showed that the parenting stress of mothers with children with CP showed a decreasing trend over time. In addition, it was suggested that the gross motor function level of the children affected the initial level of parenting stress but did not affect subsequent changes. Based on these findings, directions for further research are put forth.

## Figures and Tables

**Figure 1 healthcare-11-01317-f001:**
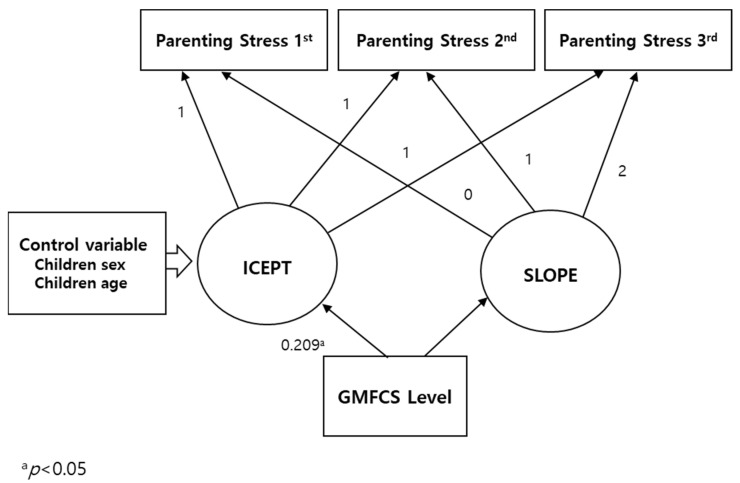
Relationship between changes in parenting stress and GMFCS level. ICEPT = initial value of parenting stress, SLOPE = change rate of parenting stress, and GMFCS = Gross Motor Function Classification System. The numbers on the lines are the regression weights.

**Table 1 healthcare-11-01317-t001:** Participants’ characteristics (children with CP and their mothers) at baseline.

Characteristics	Frequency	%	χ2
Children with CP			
Sex			
Male	90	55.6	2.00
Female	72	44.4	
GMFCS level			
Level 1	21	13.0	75.10 **
Level 2	25	15.4	
Level 3	19	11.7	
Level 4	21	13.0	
Level 5	76	46.9	
Mothers			
Age			
30~39	58	35.8	199.83 **
40~49	120	74.1	
50≤	9	5.6	
Missing value	1	0.6	
Education level			
Graduate school	20	12.3	171.04 **
University	92	56.8	
High school	44	27.2	
Middle school	2	1.2	
No response	4	2.5	
Employment			
Yes	51	31.5	37.540 **
No	107	66.0	
No response	4	2.5	

Note: GMFCS = Gross Motor Function Classification System; ** *p* < 0.01.

**Table 2 healthcare-11-01317-t002:** Descriptive statistics of parenting stress.

Category	First Measurement (*n* = 162)		Second Measurement (*n* = 162)		Third Measurement (*n* = 162)	
	M	SD	M	SD	M	SD
Sub-category						
Parental distress	34.69	8.33	32.89	7.84	33.25	7.81
PCDI	32.38	6.63	31.42	6.13	31.49	6.37
DCC	30.51	7.71	29.88	7.46	29.13	7.50
GMFCS						
Level 1	92.24	23.50	97.51	19.20	83.10	19.44
Level 2	91.79	20.76	86.14	21.76	93.42	19.93
Level 3	101.83	17.69	90.04	20.00	97.06	15.88
Level 4	92.40	14.82	100.50	16.93	97.95	19.20
Level 5	101.15	18.08	99.15	14.93	95.65	17.04
Total	97.58	19.03	94.19	18.03	93.87	18.34

Note: M = mean, SD = standard deviation, PCDI = parent–child dysfunctional interaction, DCC = difficult child characteristics, and GMFCS = Gross Motor Function Classification System.

**Table 3 healthcare-11-01317-t003:** Correlation between variables.

Category	Children’s Age	GMFCS	Parenting Stress in 1st Measurement	Parenting Stress in 2nd Measurement
GMFCS	0.275 **			
Parenting Stress in 1st measurement	0.066	0.183 *		
Parenting Stress in 2nd measurement	0.105	0.143	0.623 **	
Parenting Stress in 3rd measurement	0.128	0.128	0.606 **	0.650 **

Note: GMFCS = Gross Motor Function Classification System, * *p* < 0.05, and ** *p* < 0.01.

**Table 4 healthcare-11-01317-t004:** Model suitability of parenting stress.

Model	χ^2^	df	NFI	CFI	TLI	RMSEA
linear	2.985	3	0.984	1.000	1.000	0.000
Wave 2 variation model	0.306	3	0.998	1.000	1.015	0.000
Wave 3 variation model	8.922	3	0.952	0.968	0.968	0.111

Note: NFI = normed fit index, CFI = comparative fit index, TLI = Tucker–Lewis index, and RMSEA = root mean square error of approximation.

**Table 5 healthcare-11-01317-t005:** Estimates of the latent growth model.

Category		Mean	*p*	Variance	*p*
Parenting stress Linear model	Intercept	95.883 ***	<0.001	209.505 ***	<0.001
	Slope	−2.552 *	0.042	−14.908	0.628
	Covariance		2.437		

Note: ICEPT = initial value of parenting stress, SLOPE = change rate of parenting stress, * *p* < 0.05, and *** *p* < 0.001.

**Table 6 healthcare-11-01317-t006:** Predictors of parenting stress variation.

Category		Intercept		Slop	
		B	β	B	β
Control variable	Children’s sex	2.023	0.069	−1.948	−0.551
	Children’s age	0.088	0.028	0.139	0.970
Predictor variable	GMFCS	1.994 *	0.209	−0.161	−0.139

Note: ICEPT = initial value of parenting stress, SLOPE = change rate of parenting stress, GMFCS = Gross Motor Function Classification System, B = unstandardized estimate, β = standardized estimates, and * *p* < 0.05.

## Data Availability

The data presented in this study are available from the corresponding author upon request.

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
