# Peer review of "Longitudinal Changes in the Parenting Stress of Mothers of Children with Cerebral Palsy and Their Relationship with Children’s Gross Motor Function System Levels"

_healthcare, 2023, doi:10.3390/healthcare11091317_

Round 1

Reviewer 1 Report

Thank you for this review

It is a good article that talks about a topic that is very current and that is gaining momentum in daily clinical practice; but there are some aspects to improve

Abstract: the sections are missing, since some are used but others are not. The keywords could be improved to better adapt.

Intro: ok

Methods: except Jeju Island, why? Why are they not included before 3 years? There are already tools that speak of CP before. The average age and table 1 must go to results

Results: explain figure 1

Discussion: the objective of the study should no longer appear in the first paragraph, but rather the results that have been obtained, which are reflected in the following paragraphs, so 187 to 190 would be left over.

It could be added in future lines that it would be interesting to do the same study but between fathers and mothers, and even with older siblings; to see how it affects family functioning

Conclusions: this should be the conclusion, much shorter and more concise “The results showed that parenting 22 stress of mothers with children with cerebral palsy was decreasing over time and that Gross Motor 23 Function Classification System level was related to parenting stress level. Based on these findings, 24 a proposal for a further research is put forth.”

Regards

Author Response

Review 1

Comment #1: It is a good article that talks about a topic that is very current and that is gaining momentum in daily clinical practice; but there are some aspects to improve

Response #1: Thank for your comments. Your suggested revisions have helped me a lot to improve the manuscript.

Comment #2: The keywords could be improved to better adapt.

Response #2: Keywords have been revised according to comment.

Keywords: mothers; children with cerebral palsy; gross motor classification system; parenting stress; longitudinal study; latent growth model

Comment #2: except Jeju Island, why? Why are they not included before 3 years? There are already tools that speak of CP before.

Response #2: In the process of asking about the intention of the initial sampling, there was no expression of intention to participate in Jeju Island, so it was excluded from sampling (page 2, line 79 ~ line 81).

All mothers were primary caregivers. Sampling was done across the country, with the exception of Jeju Island because there was no expression of intention to participate.

Comment #3: The average age and table 1 must go to results

Response #3: Table 1 have been moved into results section (page 3; line 139~page 4; line 146).

Comment #4: explain figure 1

 Response #4: Figure legend have been added.

Figure 1. Relationship between the changes of parenting stress and GMFCS level. ICEPT = initial value of parenting stress and SLOPE = change rate of parenting stress. The numbers on the lines were the regression weights.

Comment #5: the objective of the study should no longer appear in the first paragraph, but rather the results that have been obtained, which are reflected in the following paragraphs, so 187 to 190 would be left over.

Response #5: First paragraph has been revised according to comment (page 6, lien 194 ~ line 200).

Mothers’ experience parenting stress during the performance of parenting roles and throughout the developmental process of the family [30]. Previous studies on mothers’ parenting stress have reported that this stress changes gradually [31, 32], but studies on such changes in mothers of children with CP are few. Parenting stress weakens mothers' interest in child rearing, leading to dysfunctional parenting behavior [33]. The focus was to provide recommendations for practices and future research directions regarding the major findings.

Comment #6: It could be added in future lines that it would be interesting to do the same study but between fathers and mothers, and even with older siblings; to see how it affects family functioning

Response #6: Suggestion about further studies have been inserted according to comment (page 8, line 267 ~ line 270).

There were some limitations in this study. First, there may be potential mediating variables related to parenting that may have affected the results. In future studies, it is necessary to investigate and verify the mediating variables, such as numbers of children in the same household and marital status.

Comment #7: this should be the conclusion, much shorter and more concise “The results showed that parenting 22 stress of mothers with children with cerebral palsy was decreasing over time and that Gross Motor 23 Function Classification System level was related to parenting stress level. Based on these findings, 24 a proposal for a further research is put forth.”

Response #7: Conclusion have been revised according to comment (page 8, line 282 ~ line 287).

This study is significant as it reports the results of longitudinal changes in the parenting stress in mothers. The results show that parenting stress of mothers with children having CP showed a decreasing trend over time. In addition, it was suggested that the gross motor function level of the children affected the initial level of parenting stress but did not affect subsequent changes. Based on these findings, a proposal for further research is put forth.

Reviewer 2 Report

This paper did a nice job to describe the parenting stress of mothers for children with CP over time and to test the relationship with GMFCS levels in Korea. I really enjoyed learning from this study and agree that this paper can provide clinicians, researchers and families with children of CP valuable information of parenting stress, especially in Korea and east Asia. I have some questions and suggestions about this paper that are listed below:

1.     Title: The title is a little confusing to me for the part of GMFCS level. I suggest changing the title to “Longitudinal Changes in Parenting Stress of Mothers of Children with Cerebral Palsy and relationship to Gross Motor Function System Levels” or just “Longitudinal Changes in Parenting Stress of Mothers of Children with Cerebral Palsy”.

Introduction:

2.     Page 2, line 62: Crnic, Gaze [16] revealed ……… Because there are more than these 2 authors, need to change to “Crnic and colleagues [16] revealed……”

Methods:

3.     In this study, only mothers are included. Considering the complex parenting styles and approaches nowadays and how these factors may have huge impact of the study results, is there any other data of potential moderator or mediator related parenting, such as whether all these mothers are primary caregivers, numbers of children in the same house hold, marriage status……for further explanation?

4.     Page 2, line 80: You mentioned the sample was from across the country, Korea, except Jeju Island. Is there any specific reason for this? Will it impact the generalization of the results? Why or why not?

5.     Page 2, line 81: The inclusion criteria needs to be edit. It is a little confusing while jumping from children and mother of children…… For example criteria 2 can just be “children ranging from 3 to 15 years old at baseline”. No need for “mothers of”.

6.     Page 2, line 82: You mentioned “mothers who responded three times during……..”. Please add these mothers responded to what (e.g. phone call of survey? Mail questionnaires? Or others?) so that audience can know the format for your data collection. Also, please report the response rate and/or drop-out rate of the study. Please include and discuss area differences, GMFCS level differences, SES differences….. in terms of the response/drop-out rate if there is any.

7.     Page 2, line 85: “The mean age of children with CP was 9.6 years ……” Please add “at baseline” at the end.

8.     Page 2-3, Table 1: Please add “at baseline” after all age data. Also the employment data if this is true.

9.     Page 3, Table 1: Does missing value here really represents “missing”? Or also includes “prefer not to answer”? Please also address how you handle the missing data and/or prefer not to answer situation.

10.  Page 3, line 97: Fix citation style “(24)” to “[24]”.

11.  Page 3, line 97: Ref 25 was a study only included sample within Korea, correct? If so, please make it clear throughout the paper, and also include the area information for other studies from western areas you cited as we know there can be big impact of parenting stress of people in and from different culture background.

12.  Page 3, line 100: About K-PSI-SF, was it designed (and the psychometric tests) only for mothers? If so, need to make it clear. If the norm actually include other parents, please also make it clear to readers.

13.  Page 3, line 105: “the entire test was reported to be .92”. I assume 0.92 is Cronbach's alpha, but please add what measure you present here.

Results:

14.  I understand this study use the short-form of the survey, but any chance you have category data to share and to initiate more detailed discussion?

15.  Page 4, Table 2: It would be very helpful to add the data by GMFCS level as well for more information and discussion.

16.  Page 4, Table 3: Please add which correlation coefficient you presented here.

17.  Page 5, Table 5: Highly suggest to just present the p values for all. The multiple stars system does not work quite clear and less information.

18.  Page 5, Table 6: Please provide information about what “B” is in this table, and please check for all the tables as well. It is very important to keep all these statistics clear.

19.  Page 6, Figure 1: Please add legend on: a. what are the numbers on the lines? B. What ICEPT and SLOPE stand for?

Discussion:

20.  For papers you cited which data was collected 15 years ago, such as papers published in 2007, it would be helpful to consider the potential time threats/difference as the parenting styles, ways, and related factors have been actually changing a lot since then.

21.  I understand that why mother is the only sample population for this study. However, I believe it is important too to bring the discussion on fathers or other format of parents into this paper to remind/suggest for further considerations.

22.  Page 6, line 191: “First, it was confirmed that……” Considering the study design, it is not appropriate to use “confirm” I highly suggest to change it to “suggest”, or “test” or “recommend”. Also, please check the entire Discussion section, especially page 6 line 190-202, as this issue shows in many places.

23.  Page 6, line 191-202: You explain the statistical significance here, which is very clear. However, it would be very helpful and bring your result to another level if you can provide data/discussion on whether this difference/change is clinical meaningful. To do this, you can think of providing the Minimal clinical important differences or if not available, minimal detectable difference of PSI-SF for comparison to your results.

24.  Page 6, line 202: Please add ”in different GMFCS level” in the end of the final sentence.

25.  Page 6, line 203-215: In this section, you discuss the similarity and difference between this study to referenced studies. Considering the sample from previous studies are actually quite different from this study. I highly recommend you to add more information of demographics that can potentially impact parenting style and stress, such as area, culture, the data collection years, social status, race etc. I list the previous examples based on my understanding of the referenced papers, but the list should not be limited as what I have here as you are surly know more about what are the factors should be discussed.

26.  Page 7, line 239-244: In this part, you mentioned the differences between your results and previous publication may be due to “the stability of GMFCS levels” and try to explain it. However, the sentences are very confusing to me that I cannot get the point. Please work on this part to make it more clear and help me and the readers to understand your rationale.

Author Response

Reviewer 2

Comment #1:  This paper did a nice job to describe the parenting stress of mothers for children with CP over time and to test the relationship with GMFCS levels in Korea. I really enjoyed learning from this study and agree that this paper can provide clinicians, researchers and families with children of CP valuable information of parenting stress, especially in Korea and east Asia. I have some questions and suggestions about this paper that are listed below:

Response #1: Thank for your comments. Your suggested revisions have helped me a lot to improve the manuscript.

Comment #2: Title: The title is a little confusing to me for the part of GMFCS level. I suggest changing the title to “Longitudinal Changes in Parenting Stress of Mothers of Children with Cerebral Palsy and relationship to Gross Motor Function System Levels” or just “Longitudinal Changes in Parenting Stress of Mothers of Children with Cerebral Palsy”.

Response #2: Title have been revised according to comment.

Longitudinal Changes in the Parenting Stress of Mothers of Children with Cerebral Palsy and its Relation to Gross Motor Function System Levels

Comment #3: Page 2, line 62: Crnic, Gaze [16] revealed ……… Because there are more than these 2 authors, need to change to “Crnic and colleagues [16] revealed……”

Response #3: It have been corrected according to comment (page 2, line 62).

Crnic et al. [16]

Comment #4: In this study, only mothers are included. Considering the complex parenting styles and approaches nowadays and how these factors may have huge impact of the study results, is there any other data of potential moderator or mediator related parenting, such as whether all these mothers are primary caregivers, numbers of children in the same house hold, marriage status……for further explanation?

Response #4: They were all primary caregivers. However other information such as numbers of children in the same household and marriage status. Concerns about potential moderator or mediator related parenting have been described in limitation of this study section (page 8, line 267 ~ line 275).

There were some limitations in this study. First, there may be potential mediating variables related to parenting that may have affected the results. In future studies, it is necessary to investigate and verify the mediating variables, such as numbers of children in the same household and marital status. This study investigated the relation between changes in parenting stress and GMFCS levels of children with CP in only mothers. Although mothers are the primary caregivers for children with CP, studies should be conducted to find out the caregiving burden of other caregivers such as fathers and older siblings. Additionally, a study should be conducted to test how their caregiving burden affects family functioning.

Comment #5: Page 2, line 80: You mentioned the sample was from across the country, Korea, except Jeju Island. Is there any specific reason for this? Will it impact the generalization of the results? Why or why not?

Response #5: In the process of asking about the intention of the initial sampling, there was no expression of intention to participate in Jeju Island, so it was excluded from sampling (page 2, line 79 ~ line 81).

All mothers were primary caregivers. Sampling was done across the country, with the exception of Jeju Island because there was no expression of intention to participate.

Comment #6: Page 2, line 81: The inclusion criteria needs to be edit. It is a little confusing while jumping from children and mother of children…… For example criteria 2 can just be “children ranging from 3 to 15 years old at baseline”. No need for “mothers of”.

Response #6: The inclusion criteria have been revised according to comment (page 2, line 81 ~ line 83).

The inclusion criteria were: 1) children with a medical CP diagnosis and 2) children ranging from 3 to 15 years of age. The participants were mothers who responded thrice during the two-year study period.  

Comment #7: Page 2, line 82: You mentioned “mothers who responded three times during……..”. Please add these mothers responded to what (e.g. phone call of survey? Mail questionnaires? Or others?) so that audience can know the format for your data collection. Also, please report the response rate and/or drop-out rate of the study. Please include and discuss area differences, GMFCS level differences, SES differences….. in terms of the response/drop-out rate if there is any.

Response #7: The differences according to general characteristics of children and their mothers have been analyzed and described. There were no drop-out (page 3, line 139 ~ line 14; Table 1).

Comment #8:  Page 2, line 85: “The mean age of children with CP was 9.6 years ……” Please add “at baseline” at the end.

Response #8: “at baseline” have been added at the end (page 3, line 140).

The mean age of the children with CP was 9.6 years (SD = 4.69) at baseline.

Comment #9: Page 2-3, Table 1: Please add “at baseline” after all age data. Also the employment data if this is true.

Response #9: “at baseline” have been added in the table title (page 3, Table 1).

Table 1. Participants’ Characteristics (children with CP and their mothers) at baseline.

Comment #10: Page 3, Table 1: Does missing value here really represents “missing”? Or also includes “prefer not to answer”? Please also address how you handle the missing data and/or prefer not to answer situation.

Response #10: Missing value presented “no response”. General characteristics did not used for analysis. “Missing” have been revised “No response”.

Comment #11: Page 3, line 97: Fix citation style “(24)” to “[24]”.

Response #11: Thank you for your correct.

Comment #12: Page 3, line 97: Ref 25 was a study only included sample within Korea, correct? If so, please make it clear throughout the paper, and also include the area information for other studies from western areas you cited as we know there can be big impact of parenting stress of people in and from different culture background.

Response #12: Area have been inserted according to comment (page 1, line 95).

Park [25] reported that the coefficients ranged from 0.690 to 0.789 and the GMFCS remained stable in children with CP aged 2 to 12 years within South Korea.

Comment #13: Page 3, line 100: About K-PSI-SF, was it designed (and the psychometric tests) only for mothers? If so, need to make it clear. If the norm actually include other parents, please also make it clear to readers.

Response #13: Mothers of children with CP have been inserted (page 3, line 102 ~ 103).

Rasch analysis of mothers having children with CP; the internal consistency (Cronbach’s α) of the entire test was reported to be .92 [26].

Comment #14: Page 3, line 105: “the entire test was reported to be .92”. I assume 0.92 is Cronbach's alpha, but please add what measure you present here.

Response #14: Cronbach’s alpha has been inserted (page 3, line 102 ~ 103).

Rasch analysis of mothers having children with CP; the internal consistency (Cronbach’s α) of the entire test was reported to be .92 [26].

Comment #15: I understand this study use the short-form of the survey, but any chance you have category data to share and to initiate more detailed discussion?

Response #15: The mean parenting stress of sub-category have been inserted in Table 2 (page 4, Table 2).

Comment #16: Page 4, Table 2: It would be very helpful to add the data by GMFCS level as well for more information and discussion.

Response #16: The data by GMFCS level have been added (page 3, Table 1; page 4, Table 2).

Comment #17: Page 4, Table 3: Please add which correlation coefficient you presented here.

Response #17: Add correlation coefficient have been presented in the manuscript (page 5, line 153 ~ line 158).

The intercorrelations among study variables are presented in Table 3. The GMFCS level and parenting stress in the first year was significantly related (p < 0.05). The coefficient was 0.623 between the first measurement of parenting stress and the second measurement. The coefficient was 0.606 between the first and third measurements. The coefficient was 0.605 between the second and third measurements. The correlation coefficient between children’s age and GMFCS was 0.275 (p < .01).

Comment # 18: Page 5, Table 5: Highly suggest to just present the p values for all. The multiple stars system does not work quite clear and less information.

Comment #18: P-value have been added (page 5, Table 5).

Comment #19: Page 5, Table 6: Please provide information about what “B” is in this table, and please check for all the tables as well. It is very important to keep all these statistics clear.

Response #19: The information about B and β have been added (page 6, Table 6).

Comment #20: Page 6, Figure 1: Please add legend on: a. what are the numbers on the lines? B. What ICEPT and SLOPE stand for?

Response #20: Figure legend have been added.

Figure 1. Relationship between the changes of parenting stress and GMFCS level. ICEPT = initial value of parenting stress and SLOPE = change rate of parenting stress. The numbers on the lines were the regression weights.

Comment #21: For papers you cited which data was collected 15 years ago, such as papers published in 2007, it would be helpful to consider the potential time threats/difference as the parenting styles, ways, and related factors have been actually changing a lot since then.

Response #21: According to comment, statement about potential time treats have been inserted (page 8, line 277 ~ line 280).

As the collection of the data used in this study began in 2007 and ended in 2009, potential time threats should be considered while interpreting the findings. Since there is a possibility that factors related to parenting style and methods have changed after data collection, an additional longitudinal study on the parenting stress of mothers of children with CP could provide implications for comparison with this study’s results.

Comment #22: I understand that why mother is the only sample population for this study. However, I believe it is important too to bring the discussion on fathers or other format of parents into this paper to remind/suggest for further considerations.

Response #22: Suggestion for further study for other family members have been inserted in discussion section (page 8, line 270 ~ line 275).

This study investigated the relation between changes in parenting stress and GMFCS levels of children with CP in only mothers. Although mothers are the primary caregivers for children with CP, studies should be conducted to find out the caregiving burden of other caregivers such as fathers and older siblings.

Comment #23: Page 6, line 191: “First, it was confirmed that……” Considering the study design, it is not appropriate to use “confirm” I highly suggest to change it to “suggest”, or “test” or “recommend”. Also, please check the entire Discussion section, especially page 6 line 190-202, as this issue shows in many places.

Response #23: The word of “confirm” have been revised throughout the manuscript (throughout manuscript).

Comment #24: Page 6, line 191-202: You explain the statistical significance here, which is very clear. However, it would be very helpful and bring your result to another level if you can provide data/discussion on whether this difference/change is clinical meaningful. To do this, you can think of providing the Minimal clinical important differences or if not available, minimal detectable difference of PSI-SF for comparison to your results.

Response #24: Clinical meaningful interpretation and the Minimal clinical important differences have been added (page 6, line 211 ~ line 215).

To provide the clinical meaning of the results, minimal clinical important differences (MCID) were calculated using Cohen’s d [35]. The d between Wave 1 and Wave 2 was .02, .0.02 between Wave 2 and 3, and 3 between Wave 1 and Wave 3. Although the MCIDs across the measurement time were small, decreasing stress in mothers of children with CP might indicate the mother’s adaptability to child rearing.

Comment #25:  Page 6, line 202: Please add ”in different GMFCS level” in the end of the final sentence.

Response #25: It have been added (page 7, line 220~ line 221).

Comment #26: Page 6, line 203-215: In this section, you discuss the similarity and difference between this study to referenced studies. Considering the sample from previous studies are actually quite different from this study. I highly recommend you to add more information of demographics that can potentially impact parenting style and stress, such as area, culture, the data collection years, social status, race etc. I list the previous examples based on my understanding of the referenced papers, but the list should not be limited as what I have here as you are surly know more about what are the factors should be discussed.

Response #26: As your comments, more detailed information related parenting stress have been added (page 7, line 223 ~ line 231).

Mulsow and Caldera [30] measured changes in parenting stress for 164 mothers and infants using random sampling and reported that the mothers' parenting stress initially increased and then decreased between two and three years of age in the USA. Williford and Calkins [31] measured changes in parenting stress of 430 mothers of children between the ages of two and five and reported that the mothers' parenting stress decreased after two years of age in the USA. The results of a longitudinal study on parenting stress for young, low-income, African-American first-time mothers (n = 120) was conducted by Chang and Fine [36], which showed that mothers’ stress gradually decreased with their children’s increasing age.

Comment #27: Page 7, line 239-244: In this part, you mentioned the differences between your results and previous publication may be due to “the stability of GMFCS levels” and try to explain it. However, the sentences are very confusing to me that I cannot get the point. Please work on this part to make it more clear and help me and the readers to understand your rationale.

Response #27: Statements have been revised to explain clearer than previous (page 7, line 256 ~ page 8, line 1).

The effect of children’s GMFCS level on the initial value in this study was in line with the findings of previous studies. However, there is a difference in the results of this study; children’s GMFCS level did not have an effect on change of parenting stress of their mothers. Mother’s parenting stress decreased regardless of the child’s level. These results are probably related to the stability of GMFCS levels in children with CP. Stability of GMFCS levels in children with CP has already been reported [41, 42]. While the GMFCS level did not change over time, the mother’s parenting stress decreased. This implies that there were factors that mitigated parenting stress.